# Determining Risk Factors Associated with Depression and Anxiety in Young Lung Cancer Patients: A Novel Optimization Algorithm

**DOI:** 10.3390/medicina57040340

**Published:** 2021-04-01

**Authors:** Yu-Wei Fang, Chieh-Yu Liu

**Affiliations:** 1Department of Nephrology, Shin Kong Memorial Wu Ho-Su Hospital, Taipei 111, Taiwan; M005916@ms.skh.org.tw; 2Department of Medicine, Fu-Jen Catholic University, New Taipei 242, Taiwan; 3Biostatistical Consulting Lab, Department of Speech Language Pathology and Audiology, National Taipei University of Nursing and Health Sciences, Taipei 112, Taiwan; 4Department of Teaching and Research, Taipei City Hospital, Taipei 106, Taiwan

**Keywords:** young lung cancer, depression, anxiety, multiple correspondence analysis, k-means clustering

## Abstract

*Background and Objectives*: Identifying risk factors associated with psychiatrist-confirmed anxiety and depression among young lung cancer patients is very difficult because the incidence and prevalence rates are obviously lower than in middle-aged or elderly patients. Due to the nature of these rare events, logistic regression may not successfully identify risk factors. Therefore, this study aimed to propose a novel algorithm for solving this problem. *Materials and Methods*: A total of 1022 young lung cancer patients (aged 20–39 years) were selected from the National Health Insurance Research Database in Taiwan. A novel algorithm that incorporated a *k*-means clustering method with *v*-fold cross-validation into multiple correspondence analyses was proposed to optimally determine the risk factors associated with the depression and anxiety of young lung cancer patients. *Results*: Five clusters were optimally determined by the novel algorithm proposed in this study. *Conclusions*: The novel Multiple Correspondence Analysis–*k*-means (MCA–*k*-means) clustering algorithm in this study successfully identified risk factors associated with anxiety and depression, which are considered rare events in young patients with lung cancer. The clinical implications of this study suggest that psychiatrists need to be involved at the early stage of initial diagnose with lung cancer for young patients and provide adequate prescriptions of antipsychotic medications for young patients with lung cancer.

## 1. Introduction

Lung cancer is a very aggressive malignant disease; people who smoke or are exposed to polluted environments or with genetic mutation may be at significantly higher risk of lung cancer [1,2,3]. Published studies have shown that Asian women who never smoke still have a higher risk of lung cancer compared with women in European countries or the United States [4,5]. Based on a worldwide report issued by the International Agency for Research on Cancer in 2018, for both males and females, lung cancer had become the most prevalent cancer globally (with incidence rate of 11.6% of all cancers) and had been ranked as the leading cause of cancer death (death rate of 18.4% of the total cancer deaths). The economic burden of treatments and care for lung cancer has also globally increased in recent years [6,7]. Published studies have showed that lung cancer is significantly associated with older age (70 years old being the average age of initial diagnosis) [7,8], but very low incidence rate in young people aged 20–40 years around the world [8]. In recent decades, due to the dramatic improvements of clinical treatments and screening techniques for lung cancer, the survival of newly diagnosed lung cancer patients has been significantly prolonged [9,10] and the incidence rate in young patients with lung cancer also increased [1,5,8]. Published studies also showed that young patients with lung cancer had better treatment outcomes of receiving surgery, chemotherapy, or radiotherapy and have relatively longer relapse-free survival, which indicates that young lung cancer patients are more likely to have prolonged survival [11,12,13].

Therefore, young patients with lung cancer are a noteworthy group of patients, because they have obviously lower incidence rate and prevalence rate than middle-age or elder people and they may have longer survival time. Liu et al. (2019) [14] used a retrospective review of patients with lung cancer in one hospital in China from January 2010 to June 2017, the prevalence of lung cancer in young adults aged between 18 and 35 years old was 1.37%; and Rich et al. (2015) [15] also used a retrospective cohort review using a validated national audit dataset and the results showed that the prevalence of lung cancer in young adults aged between 18 and 39 years was 0.5%. The overall incidence and prevalence in elder age groups (>50 years old) was increasing in recent decade, however, the incidence and prevalence of lung cancer in young adults (<40 years old) can be still regarded as relatively low in nowadays global cancer epidemiology. Recent studies showed that young lung cancer survivors are also at a high risk of psychiatric diseases, such as anxiety and depression in the following years of survival [16,17,18]. However, most of the published studies used self-reported scales or questionnaires to measure anxiety and depression instead of using diagnoses by psychiatrists; therefore, the so-called depression or anxiety in published studies can solely regarded as depression symptoms or anxiety symptoms. For example, Yan et al. [17] showed that the anxiety and depression prevalence rates of lung cancer patients were 43.5% and 57.1% by using the Hospital Anxiety and Depression Scale (HADS), which look high proportions in lung cancer patients. In addition, if young lung cancer patients who may have prolonged survival and are at high risk of psychiatrist-confirmed depression and anxiety, they will consume considerable medical resources due to the additional treatments for psychiatric diseases [6,19]. Nevertheless, there is still a lack of literature investigating risk factors associated with psychiatrist-confirmed depression and anxiety in young lung cancer patients. This study was aimed to develop a novel algorithm for identifying risk factors for psychiatrist-confirmed anxiety and depression in young lung cancer patients aged 20–39 years old by using the population-based database (National Health Insurance Research Database (NHIRD) in Taiwan), which can assist clinicians or young patients with lung cancer in preventing anxiety and depression at early stages.

## 2. Materials and Methods

### 2.1. Study Design and Study Database

This study design of this research adopted the secondary analysis of longitudinal data from NHIRD. The study database used here was retrieved from the NHIRD in Taiwan. Since the National Health Insurance (NHI) program was launched on 1 March 1995, the NHI program provided healthcare service coverage to more than 99% of the population by 2017 [20]. The NHIRD includes medical reimbursement records for outpatient and inpatient healthcare services, hospital or clinic visits, dental service visits and traditional Chinese medicine service visits. All of the reimbursement records for diagnostic and medical-related procedures for diseases are based on the international classification of diseases (ICD)—ninth and tenth revisions (after 1 January 2016 [21]) of the clinical modification (CM, or ICD-9-CM and ICD-10-CM, respectively)—and on a procedure coding system for all medical service claims.

### 2.2. Ethics Statement

The ethical review of this study was approved by the Institutional Review Board of the School of Nursing, National Taipei University of Nursing and Health Sciences (approval number: IRB# CN-IRB-2011-063). The date of approval was 23 October 2011. The encryption and protection of the personal information from the NHIRD were performed by the National Health Insurance Administration in Taiwan by using a complex double encryption procedure. In addition, because the present study was a secondary data analysis, written informed consent forms were not required from the recruited or selected patients. This study was also registered at Open Science Framework (OSF, reference osf.io/fkhm8 (accessed on 15 March 2021)).

### 2.3. Study Population and Possible Risk Factors Selection

The ICD-9-CM codes that were used to define patients with depression were 296.2X–296.3X, 300.4 and 311.X and the ICD-9-CM codes used to define patients with anxiety were 300.XX, 291.89 and 292.89. In Taiwan, if cancer patients are suspected of having depression or anxiety, they are refereed by the oncologists to psychiatrists, which is recorded as the first National Health Insurance (NHI) outpatient visit. After the referral, the cancer patients receive some psychological tests by clinical psychologists and the cancer patients are diagnosed by psychiatrists again to determine if they need anti-depressant or anti-anxiety medications; this is recorded as the second NHI psychiatric visit. After a period of time, the cancer patients need to be confirmed again by psychiatrists; therefore, to confirm that a cancer patient has depression or anxiety usually needs at least three outpatient visits and the prescription of anti-depressant or anti-anxiety drugs. In this study, young lung cancer patients that were aged 20–39 years and who were newly diagnosed with lung cancer (ICD-9-CM code = 162.XX) between 1 January 2001, and 31 December 2007, were retrieved from the NHIRD. Young lung cancer patients who died or withdrew from the NHI program during the study period were excluded. Young patients with lung cancer who had been diagnosed with baseline psychiatric diseases, such as depressive disorder (ICD-9-CM codes: 296.2X–296.3X, 300.4 and 311.X), anxiety states (ICD-9-CM codes: 300.XX, 291.89 and 292.89), bipolar disorders (ICD-9-CM codes: 296.0, 296.1, 296.4, 296.5, 296.6, 296.7, 296.8, 296.80 and 296.89), or alcohol-induced mental disorders (ICD-9-CM codes: V113, 9800, 2650, 2651, 3575, 4255, 3050, 291, 303 and 571.0–571.3) between 1 January and 31 December in 2001 were also excluded. In order to avoid selecting false-positive patients with depression and anxiety, young lung cancer patients with at least three consecutive corresponding diagnoses were eligible to be coded as having depression and anxiety.

The possible risk factors associated with depression and anxiety among lung cancer patient were determined based on Park et al. [19], who investigated if hypertension, diabetes mellitus, history of tuberculosis, liver disease (liver cancer and liver cirrhosis), end-stage renal disease, coronary artery disease (including heart failure), stroke (ischemic stroke and hemorrhage stroke) and Chronic obstructive pulmonary disease (COPD) are risk factors associated with anxiety and depression after surgical treatment for lung cancer; and Clarke and Currie [20], who took into account heart disease, stroke, cancer, diabetes mellitus, rheumatoid arthritis and asthma as the possible risk factors associated with depression and anxiety in cancer patients. Therefore, in this study, we took into account diabetes mellitus (DM), hypertension, asthma, liver cirrhosis, COPD, autoimmune diseases (including rheumatoid arthritis, systemic lupus erythematosus and aplastic anemia), cerebral diseases (including ischemic stroke, hemorrhage stroke and transient ischemic attack (TIA)), heart failure, hepatitis B virus (HBV), renal diseases and osteoporosis.

### 2.4. Combining Multiple Correspondence Analysis and the K-Means Clustering Algorithm with v-Fold Cross-Validation (MCA–k-Means Clustering Algorithm)

The raw data matrix was first transformed into a matrix with solely index variables (i.e., encoded as 0 or 1) through multiple correspondence analysis (MCA) [21,22], which was the data preprocessing procedure for the raw data matrix. The index variables indicate the levels of all of the categorical variables in this study. The MCA then converted all index variables into multi-dimensional Euclidean coordinates. The multi-dimensional Euclidean coordinate matrix derived from the MCA could be considered a high-dimensional dataset that could be carried into the further optimal clustering algorithm. In order to determine the optimal clustering in the high-dimensional dataset obtained from the MCA, the k-means clustering algorithm with *v*-fold cross-validation was applied to obtain the optimal clustering. The algorithm is described in detailed in the following:

#### 2.4.1. Step 1. Multiple Correspondence Analysis

Let **M**_I×K_ be the raw data matrix with I subjects and k categorical variables.

(1)Transform the raw data matrix into a Burt matrix:
If a categorical variable is binary, then place it in the Burt matrix as an original variable matrix.If a categorical variable has more than two levels (i.e., J_k_ > 2 levels), then convert this variable into an index variable (containing only 0 and 1); this forms an indicator matrix I × J_k_ where each column contains index variables coded with 0 or 1.Place all index variable columns together to form the indicator matrix **X**_I×J_.Calculate the Burt matrix as (**X**_I×J_)**′****·X**_I×J_.

(2)Calculate the column and row coordinates as follows:
The total orders of M_I×K_ (N) are observed and the probability matrix is defined as P = N − 1X.Define r as the vector of the row totals of P (i.e., r = P1, where 1 is a unit vector of ones) and define c as the vector of the column totals of P. Then, Dc = diag{c} and Dr = diag{r}.Calculate the Euclidean coordinates by using a singular value decomposition method as follows:
$$D_{r}^{-\frac{1}{2}}\left(Z-rc^{T}\right)D_{c}^{-\frac{1}{2}}=P\Delta Q^{T}$$
where Δ and $\Lambda =\Delta^{2}$ are the diagonal matrix of singular values and the matrix containing eigenvalues, respectively. Therefore, the row and column coordinate matrices (**F** and **G**, respectively) are calculated as follows:$$F=D_{r}^{-\frac{1}{2}}P\Delta$$
$$G=D_{c}^{-\frac{1}{2}}Q\Delta$$

(3)The number of dimensions is determined using an inertia value as follows:
The inertia value is calculated based on a Pearson chi-squared ($\chi^{2})$ value from the rows and columns to identify their coordinate centers as follows:$$d_{r}=diag\left\{FF^{T}\right\}\;\mathrm{and}\;d_{c}=diag\left\{GG^{T}\right\}.$$If a subset of **F** or **G** is selected, then the inertia values for the row and column coordinates are calculated as:
$$Inertia_{r}=\frac{diag\left\{F{F^{\prime }}^{T}\right\}}{N}\;\mathrm{and}\;Inertia_{c}=\frac{diag\left\{G{G^{\prime }}^{T}\right\}}{N},$$
where **F**′ and **G**′ are subsets of **F** and **G**.

#### 2.4.2. Step 2. K-Means Clustering with v-Fold Cross-Validation

The *k*-means clustering algorithm with *v*-fold cross-validation was applied to analyze the **F** and **G** that were obtained from the MCA [23,24]. The algorithm is as follows:(1)Determination of the range of numbers of clusters for the *k*-means clustering algorithm: In this study, the number was set from *k* = 2 to *n*, where *n* ≤ 10;(2)Determination of the initial cluster centers: The initial cluster centers were selected at random;(3)Iteration scheme: Assigning all index variables to their nearest cluster centers. The Euclidean distance was used as the distance measurement in the iterative classification scheme;(4)To determine the optimal clustering, *v*-fold cross-validation was applied to estimate the optimal number of clusters and the optimal clustering. The details of the *v*-fold cross-validation are as follows:
(a)Divide **F** or **G** into *v* folds (denoted F_i_ or G_i_, I = 1, …, *v*), in this study, we set *v* = 5;(b)For i = 1 to *v,* take F_i_ or G_i_ as the testing set and {**F**}\F_i_ or {**G**}\G_i_ as the training sets;(c)Compute the mean Euclidean distances, which are called the clustering costs in this study, within each cluster of training sets, set these as the new cluster centers and replace the cluster centers of the previous step;(d)Compute the mean Euclidean distances of each index variable (or the level of all of the categorical variables) of the testing set from the new cluster centers derived from the training sets;(5)Iterate from (1);(6)If *k* = ***j,*** which indicates the minimum mean Euclidean distances (i.e., minimal clustering cost) of each index variable of the testing set, ***j*** would be the optimum number of clusters.(7)Clustering stopping rule: If |D¯j+1−D¯j| < 0.01, then stop further dividing and clustering.(8)Regarding the determination of number of clusters, we adopted the method proposed by Wang [25], the optimal algorithm will iterate in order to classify factors into different numbers of clusters, calculate the cluster cost (in this study, we used the mean sum of squares within clusters as the cluster cost measurement) and compare the sums of squares between clusters. If the sum of squares of *k* clusters did not show statistically significant difference from *k +* 1 clusters, the optimal number of clusters is determined as *k*.

The MY Structured Query Language (MySQL) was used for selection, linkage, processing and cleaning of the dataset from the NHIRD. The algorithm we proposed in this study was implemented with STATISTICA Data Miner ver. 10.0 (StatSoft, Inc., Tulsa, OK, USA).

## 3. Results

In the present study, 1022 young lung cancer patients aged 20–39 years were studied and their demographic information is shown in Table 1. The study sample comprised 520 male (50.9%) and 502 female patients (49.1%); 154 of the patients were aged 20–29 years old (15.1%) and 868 patients were aged 30–39 years old (84.9%).

As a result of the k-means clustering of **F** and **G,** which were Euclidean coordinate matrixes derived from the multiple correspondence analysis (MCA) and by using *v*-fold cross-validation, the clustering costs of different numbers used for the k-means clustering algorithm are shown in Figure 1. According to the results shown in Figure 1, on the basis of the clustering cost, there was no statistically significant difference between using five clusters or six clusters. Based on the principal of parsimony of clustering, the optimum number of clusters was determined to be five. Table 2 presents the clustering results that comprise these five clusters. Table 2 indicates that anxiety was clustered with osteoporosis and depression was clustered with the lack of diabetes mellitus (DM), Charlson comorbidity index (CCI) = 0 and female sex.

In addition, in the present study, a control arm statistical analysis was also performed using a multiple logistic regression model, which is the most widely used method for investigating risk factors associated with diseases. Table 3a shows the results of score tests of both dependent variables—depression and anxiety—for each independent variable. In Table 3a, no statistical significance was observed for any of the independent variables with these two dependent variables, indicating that using a stepwise variable selection strategy (forward or backward variable selection) cannot be used to find any statistically significant predictors. Furthermore, Table 3b shows the results of the multiple logistic regression model (without variable selection procedures), which also indicated that there were no statistically significant predictors (except for constant terms for both dependent variables).

## 4. Discussion

The objective of this study aimed to develop a novel algorithm for identifying risk factors for anxiety and depression in young lung cancer patients aged 20–39 years by using the population-based database (National Health Insurance Research Database (NHIRD) in Taiwan), which are regarded rare events and very limited number of methods were proposed to solve this problem. A novel algorithm was proposed in this study which integrated *v*-fold cross-validation into MCA–k-means clustering for solving the problem of determining risk factors associated with rare events.

Compared with the results of a univariate analysis using traditional multiple logistic regression analysis, which is a widely used method for determining risk factors associated with diseases (see Table 3), the results showed that none of the risk factors were statistically significantly associated with anxiety and depression, respectively, in young patients with lung cancer. Moreover, some parameter estimates were very unreliable because of their large standard errors (even bigger than the parameter estimates). In Table 3a, for the depression outcome variable, CCI = 1 vs. CCI = 0, DM, asthma, liver cirrhosis, autoimmune diseases, cerebral diseases, heart failure, renal diseases and osteoporosis indicated that parameter estimates were unreliable and exhibited extremely low odds ratios (ORs); for the anxiety outcome variable, CCI ≥ 2 vs. CCI = 0, DM, hypertension, asthma, liver cirrhosis, chronic obstructive pulmonary disease (COPD), autoimmune diseases, cerebral diseases, heart failure, hepatitis B (HBV) and renal diseases also indicated that the parameter estimates were unreliable and exhibited extremely low odds ratios (ORs), or an extremely high OR for asthma. Previous studies have indicated that parameter estimation methods such as maximum likelihood estimation provide biased or inestimable estimates for rare events [26,27]. According to King and Zeng (2001) [28], logistic regression would sharply underestimate the probability of rare events. For resolving the problems, some methods have been proposed, but there is still a lack of optimal methods and agreements on how to better estimate the coefficient of logistic regression for rare event data. In this study, not only were the dependent variables (depression and anxiety) rare events, but so were the independent variables, which may have resulted in many zeros in the database and the estimation of the standard error may have been biased. The novel algorithm proposed in this study can be considered to be a good approach for resolving rare event problems. In addition, compared with the results using self-reported questionnaire or inventory, such as Yan et al. [17], which used binary logistic regression analysis and the results showed that the risk factors of both anxiety and depression were lack of surgery and age; however, binary logistic regression did not successfully identify statistically significant risk factors in this study and the difference can be resulted from different operational definitions of depression and anxiety. Both kinds of studies using self-reported questionnaires or ICD-9-CM codes by psychiatrist-confirmed diagnoses provide different contributions to the clinical practices. Studies using self-reported questionnaires or inventory to measure depression and anxiety are more likely to look for factors associated with the self-perceived depression symptom and anxiety symptoms, which may be easier to express by patients themselves and some behavior interventions may be suggested, such as exercise, focus group consultant or health promotion life adjustment. However, the results of the current study using ICD-9-CM codes of depression and anxiety which are confirmed by psychiatrists, what young patients with lung cancer need are not only behavior interventions, but also the prescriptions of antidepressant drugs or anti-anxiety drugs, or the psychiatric hospitalization.

The advantages of the MCA–k-means clustering algorithm proposed in this study are: (1) the adoption of the clustering-based method to determine risk factors associated with rare events, which may avoid the parameter estimation problems encountered when using conventional logistic regression models; (2) the algorithm can take more than one dependent variable (≥2) into account simultaneously, especially for easily confused diseases, for example, anxiety and depression in this study. In comparison with a logistic regression model, it deals with only one dependent variable at a time. (3) The algorithm determines the optimum number of clusters by using the *v*-fold cross-validation algorithm; through the repeated random sub-sampling scheme, all observations were used for both the training and validation sets and each observation was used for validation exactly once, which can help determine the optimum number of clusters with less influence from rare event data, such as the dataset used in this study.

Regarding the final clustering results of this study (see Table 2), the results indicated that anxiety was clustered with osteoporosis and depression was clustered with the lack of DM, CCI = 0 and female sex in young patients with lung cancer. These factors were optimally clustered with anxiety and depression. The results obtained in this study are validated by other studies that have indicated that patients with anxiety and osteoporosis easily encounter more complications than those with several other disease groups [29,30,31]. The results of this study indicate that young patients with lung cancer and osteoporosis are also at a high risk for the onset of anxiety. In addition, young female lung cancer patients were also at a higher risk of the onset of depression. Previously published studies have shown that female cancer patients are at significantly higher risk of depression than males [29,32,33]. In this study, the clustering results also supported that young female lung cancer patients were at a higher risk of the onset of depression.

This study still had some limitations. First, although the National Health Insurance (NHI) program in Taiwan covers more than 98% of the Taiwanese population [34,35,36], the NHIRD does not provide information about some potential confounding factors, such as smoking, alcohol consumption, exercise habits, diet and lifestyle, which may also influence the association with the risk of anxiety and depression. Second, some young lung cancer patients who experience anxiety and depression may not consult psychiatrists; they usually express their concerns about their cancer diseases to their oncologists and the oncologists may easily neglect or ignore their patients’ anxiety and depression symptoms. Thus, cancer patients may search for religious help or may isolate themselves from people or medical professionals; therefore, the number of patients with anxiety and depression may be underestimated. Third, because the young patients with lung cancer enrolled in this study were primarily of the Chinese or Han ethnicities, the results derived from the novel algorithm proposed here require further examination and validation for generalization to other ethnicities. Furthermore, according to Lu et al. (2019) [37], in recent decades, the overall incidence of lung cancer initially increased and then gradually decreased. The surgical rate and radiotherapy rate for lung cancer showed a general downward trend, while the chemotherapy rate experienced a significantly increasing trend [30]. Although the five-year relative survival rate has increased over the years, it has remained very low for the last 20 years [31]. Therefore, this study, which used a nationwide database from 2001 to 2007, can still provide useful findings for clinicians.

## 5. Conclusions

The novel MCA–k-means clustering algorithm in this study successfully identified risk factors associated with anxiety and depression, which are considered rare events in young patients with lung cancer. The clinical implications of this study suggest that psychiatrists need to be involved at the early stage of initial diagnose with lung cancer for young patients and provide adequate prescriptions of antipsychotic medications for young patients with lung cancer.

## Figures and Tables

**Figure 1 medicina-57-00340-f001:**
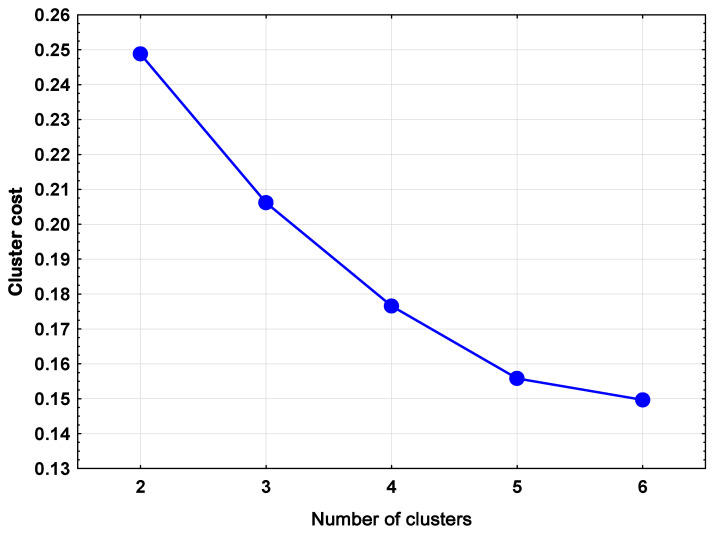
Cluster costs of different numbers of clusters resulting from k-means clustering combined with *v*-fold cross-validation.

**Table 1 medicina-57-00340-t001:** Demographic information of the study sample (*n* = 1022).

Variable	*n*	(%)
Sex		
Female	502	49.1
Male	520	50.9
Age		
20–29 y	154	15.1
30–39 y	868	84.9
Charlson comorbidity index (CCI)		
CCI = 0	870	85.1
CCI = 1	91	8.9
CCI ≥ 2	61	6
Diabetes mellitus (DM)		
Yes	23	2.3
No	999	97.7
Hypertension		
Yes	23	2.3
No	999	97.7
Asthma		
Yes	16	1.6
No	1006	98.4
Liver cirrhosis		
Yes	9	0.9
No	1013	99.1
Chronic obstructive pulmonary disease (COPD)		
Yes	51	5
No	971	95
Autoimmune diseases		
Yes	8	0.8
No	1014	99.2
Cerebral diseases		
Yes	11	1.1
No	1011	98.9
Heart failure		
Yes	2	0.2
No	1020	99.8
Hepatitis B virus (HBV)		
Yes	34	3.3
No	988	96.7
Renal diseases		
Yes	6	0.6
No	1016	99.4
Osteoporosis		
Yes	16	1.6
No	1006	98.4
Depression		
Yes	25	2.4
No	997	97.6
Anxiety		
Yes	15	1.5
No	1007	98.5

**Table 2 medicina-57-00340-t002:** Results of the multiple correspondence analysis (MCA) and k-means algorithm with *v*-fold cross-validation.

Variable	Final Classification
Autoimmune disease = Yes	1
Cerebral disease = Yes	1
Heart failure = Yes	1
Osteoporosis = Yes	2
Anxiety = Yes	2
Depression = Yes	3
DM = No	3
Age: 20–29 y	3
Age: 30–39 y	3
CCI = 0	3
Sex = Female	3
DM = Yes	4
Hypertension = Yes	4
Asthma = Yes	4
Liver cirrhosis = Yes	4
COPD = Yes	4
HBV = Yes	4
CCI ≥ 2	4
Depression = No	5
Hypertension = No	5
Asthma = No	5
Liver cirrhosis = No	5
COPD = No	5
Autoimmune disease = No	5
Cerebral disease = No	5
Heart failure = No	5
HBV = No	5
Osteoporosis = No	5
Anxiety = No	5
CCI = 1	5
Sex = Male	5

Note: DM = Diabetes mellitus; CCI = Charlson comorbidity index; COPD = Chronic obstructive pulmonary disease; HBV = Hepatitis B virus.

**Table 3 medicina-57-00340-t003:** (**a**) Score test results for each variable of the logistic regression model. (**b**) Results of multiple logistic regression models for depression and anxiety.

**(a)**								
**Variable**	**DV = Depression**				**DV = Anxiety**			
	**Score**			***p*-Value**	**Score**			***p*-Value**
Sex: Male vs. Female	0.485			0.486	0.108			0.742
Age: 30–39 vs. 20–29 years	0.017			0.895	0.036			0.85
CCI = 1 vs. CCI = 0	2.505			0.113	0.368			0.544
CCI ≥ 2 vs. CCI = 0	1.661			0.197	0.966			0.326
DM: Yes vs. No	0.59			0.442	0.35			0.554
Hypertension: Yes vs. No	0.357			0.55	0.35			0.554
Asthma: Yes vs. No	0.408			0.523	2.571			0.109
Liver cirrhosis: Yes vs. No	0.228			0.633	0.135			0.713
COPD: Yes vs. No	0.053			0.818	0.09			0.764
Autoimmune: Yes vs. No	0.202			0.653	0.12			0.729
Cerebral diseases: Yes vs. No	0.279			0.597	0.166			0.684
Heart failure: Yes vs. No	0.05			0.823	0.03			0.863
HBV: Yes vs. No	1.74			0.187	0.524			0.469
Renal diseases: Yes vs. No	0.151			0.697	0.09			0.764
Osteoporosis: Yes vs. No	0.408			0.523	2.571			0.109
**(b)**								
**Variable**	**DV = Depression**				**DV = Anxiety**			
	**Beta**	**S.E.**	**Odds Ratio (OR)**	***p*-value**	**Beta**	**S.E.**	**Odds Ratio (OR)**	***p*-value**
Sex: Male vs. Female	−0.352	0.418	0.703	0.399	−0.122	0.530	0.885	0.818
Age: 30–39 vs. 20–29 years	−0.010	0.561	0.990	0.986	0.038	0.781	1.038	0.961
CCI = 1 vs. CCI = 0	−17.329	4055.844	<0.001	0.997	0.302	0.872	1.353	0.729
CCI ≥ 2 vs. CCI = 0	1.377	0.829	3.964	0.097	−15.014	4327.707	<0.001	0.997
DM: Yes vs. No	−18.980	7844.428	<0.001	0.998	−14.048	6665.142	<0.001	0.998
Hypertension: Yes vs. No	1.217	1.092	3.377	0.265	−16.095	7460.922	<0.001	0.998
Asthma: Yes vs. No	−17.069	8269.606	<0.001	0.998	18.338	6580.883	92,044,936.212	0.998
Liver cirrhosis: Yes vs. No	−16.960	11,705.571	<0.001	0.999	−16.057	11,754.148	<0.001	0.999
COPD: Yes vs. No	−0.177	1.116	0.838	0.874	−16.910	6580.883	<0.001	0.998
Autoimmune: Yes vs. No	−17.528	12,892.859	<0.001	0.999	−17.150	13,490.401	<0.001	0.999
Cerebral diseases: Yes vs. No	−17.579	11,033.665	<0.001	0.999	−16.307	11,052.028	<0.001	0.999
Heart failure: Yes vs. No	−18.191	25,475.907	<0.001	0.999	−16.436	26,494.679	<0.001	1.000
HBV: Yes vs. No	0.225	0.962	1.252	0.815	−16.248	6243.383	<0.001	0.998
Renal diseases: Yes vs. No	−18.808	16,186.569	<0.001	0.999	−14.876	14,129.940	<0.001	0.999
Osteoporosis: Yes vs. No	−17.344	9805.003	<0.001	0.999	1.507	1.131	4.511	0.183
Constant	−3.468	0.561	0.031	<0.001	−4.152	0.778	0.016	<0.001

Note: S.E. = Standard Error; DM = Diabetes mellitus; CCI = Charlson comorbidity index; COPD = Chronic obstructive pulmonary disease; HBV = Hepatitis B virus.

## Data Availability

The study dataset (NHIRD) was not publicly archived; to access it, an application from the Bureau of National Health Insurance in Taiwan is needed. The application website is: https://www.nhi.gov.tw (the access date was 10 December 2012).

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
