# Peer review of "Determining Risk Factors Associated with Depression and Anxiety in Young Lung Cancer Patients: A Novel Optimization Algorithm"

_medicina, 2021, doi:10.3390/medicina57040340_

Round 1

Reviewer 1 Report

I think that the article in its current state is good.

Author Response

Many thanks for your valuable comments.

Reviewer 2 Report

I appreciate the authors' endeavor to improve the manuscript. 

However, I believe the following points need to be considered by the authors. 

1. Importance or value of using the novel algorithm among lung cancer patients 

The authors explained that using the novel algorithm is important since anxiety and depression among young lung cancer patients are rare events so that the method is more suitable than using logistic regression. If the reason why the authors used the novel algorithm was because they wanted to focus on rare events, it would be more appropriate to choose other cancer types that show low incidence. The authors themselves mentioned in the introduction section that "recent studies have revealed that young survivors of lung cancer are also at a HIGH risk of psychiatric diseases,"

2. Operational definition of anxiety and depression 

If the authors believe that defining anxiety and depression through clinical diagnosis is more appropriate than survey (which I also believe is more appropriate), and if they believe that this is the strength of this study, it can be whole another story. While previous researches that defined anxiety and depression through survey determined certain risk factors using logistic regression, this study showed no significant predictors using logistic regression, and this may be caused by the difference in the way of defining anxiety and depression. 

3. selecting comorbidities

Although the authors are dealing with risk factors associated with anxiety and depression among young lung cancer patients, they only focused on comorbidities.

see my previous comments point #14 and 19. 

Author Response

Many thanks for your valuable comments, please find the point-to-point reply in the attachment. Thank you!!

Reviewer 3 Report

Comments to the authors

The study provides scientific evidence that has not been published before. Its subject is interesting, it is new and will contribute to possible improvements of clinical practice. However, it needs improvements. In general, English and writing needs reviewing.

The article does not specify the registration of the study, this should be done before conducting the study and before publishing.

The abstract needs improving. The results should give more detail about the findings of the study. The conclusion should be a summary of the results and should be more concise.

The discussion section is difficult to read, it needs rewriting.

Specific comments

Introduction

Lines 51 and 53: please do not repeat However, use a different word to avoid repetition.

Line 57: ‘…but the patients did not any receive antipsychotic medications’. Please correct this phrase

Lines 58-60: This phrase needs to be corrected grammatically. ‘For example, Jacek PolaÅ„ski et al. [15] showed that among 180 lung cancer patients, the proportion of anxiety was found to be 37.2% and that of…’ Please separate ideas in different phrases. Showed that …and …was found to be.. need to be corrected.

Line 62: please avoid repetition of information: ‘almost all published studies found the prevalence of depression and anxiety using self-reported outcomes [14-16] was already mentioned in line 53-54.

Lines 62-66. This sentence is very long, please separate ideas and be concise.

Material and methods

The design of the study should be detailed at the beginning of this section.

Please clarify and specify in the manuscript when the study was conducted, the dates are not clear.

Line 118: this is part of the results

Results

Line 207: this sentence seems to be more part of the discussion or the methods.

Discussion

The first sentence is not clear, please re-write

Please review the English and shorten the sentences to facilitate reading and understanding.

The first paragraph should summarize the objective and the main findings of the study, please write accordingly.

Line 290: ‘The results obtained in this study are 290 validated by other studies that have indicated that patients with anxiety and osteoporosis 291 easily encounter more complications than those with several other disease groups [25].’ Please provide more references that support this. You refer to other studies but refer only to one.

Please provide clinical implications in the discussion section

Conclusions

The conclusions should answer the objective of the study and summarise the results obtained. Could include clinical implications but must be concise. Please re-write.

Author Response

Many thanks for your valuable comments, please find the point-to-point reply in the attachment, thank you!!

Round 2

Reviewer 2 Report

Thank you for your hard-working and specific responses. 

I would like to appreciate the fact that you've taken my requests into consideration. 

Kind regards, 

Reviewer 3 Report

Thank you for making the mofications required

This manuscript is a resubmission of an earlier submission. The following is a list of the peer review reports and author responses from that submission.

Round 1

Reviewer 1 Report

Thank you for inviting me to review this manuscript.

In this study, 1.022 young patients with lung cancer aged 20–39 years, who were newly diagnosed with HCC between January 1, 2001, and December 86 31, 2007, were retrieved from the NHIRD. This study was aimed to propose a novel algorithm: Identifying risk factors associated with anxiety and depression among young lung cancer patients.

I'm worried about is that the study is from 2001-2007, and it's 2021, 14 years later. I think it's been a long time since data collection.

 The authors affirm “The novel algorithm proposed in this study can optimally determine risk factors associated with anxiety and depression in young lung cancer patients, which may not be well identified by using conventional multiple logistic regression analysis”, but I think that is not correct. The best method to analyze this is binary logistic regression analysis. Moreover, some parameter estimates were very unreliable because of large standard errors. The logistic regression analyzes are not significant. I have not seen the mode of input of variable and data nor the percentages classified correctly by the model, nor the prediction formula of the logistic regression analysis.I think that the most correct solution was to adequately present the logistic regression model and compare it with the 5 cluster determined by the novel algorithm proposed in this study.

The authors do not provide the information about some potential risks factors such as smoking, alcohol consumption, exercise habits, diet and lifestyle, which may also influence the association with the risk of anxiety, depression and lung cancer.

Some methodological aspects could be corrected and presented better and it is an interesting proposal, however, the research data collection has been carried out between 2001 and 2007, 14 years ago.

Minor points

  1. The authors repeat the same ideas many times. For example: “By using typical logistic regression (the most 14 widely used method to identify risk factors among diseased populations)”.
  2. The format of the table is not suitable. For example, table 3a and 3b have a different format.
  3. The abbreviations used in the table must be given their meanings at the foot of the table.

Author Response

Please see the point-to-point reply in the attachment, thank you!!

Reviewer 2 Report

In this manuscript, Fang and colleagues came up with a novel algorithm using k-means clustering method with v-fold cross validation to determine risk factors associated with depression and anxiety among young patients with lung cancer. 

Major comments:

Introduction

  • First of all, as far as I'm concerned, almost all of young patients with any type of cancer would suffer from mental health problems. Even if you consider its high morality rate and growing incidence (but I am not quite sure that it is true that the recent incidence of lung cancer is actually growing in every countries. see my next comment), I don't see why you specifically targeted lung cancer patients in the study. As you mentioned in Page 2, lines #55-57, preventing psychiatric diseases among young patients at early stage is very important, and I believe this is true regardless of cancer types. The reason why you chose lung cancer patients only should be well-described in the introduction section.
  • Page 1, lines #38-39 require a reference. 
  • I don't quite get what you meant by "young patients with lung cancer who successfully control their lung cancer are more likely to have prolonged survival." (lines #41-43) What do you mean by 'who successfully CONTROL their lung cancer'? You meant "those who got appropriate medical treatments" or "those who could afford medical treatments" or something else?
  • Page 2, lines #58-59: it would be better if you could provide specific statistics. 
  • It would be helpful to understand the importance of this study if you could explain why a development of a novel algorithm is important in the introduction section. You can use some descriptions from your discussion. 

Materials and Methods 

  • Page 2, line #86: what's the abbreviation HCC stands for? It's not hepatocellular carcinoma, is it? 
  • Page 3, line #1: "2002" Is it a typo? Shouldn't it be 2001?
  • What's the code(s) that you used to define patients with depression? What's the code(s) that you used to define patients with anxiety? How are patients usually diagnosed with depression and anxiety in Taiwan? 
  • The date of IRB approval was Oct 2011 and the study period is from 2001 to 2007. Just out of curiosity, what made you rule out those medical record from 2008 to 2010?
  • What's the survival status of study population at the time of data collection? 
  • The final number of study population should be described in the material and methods section. 
  • since the purpose of this study is to come up with a new algorithm, I guess descriptions regarding MCA-k-means clustering algorithm - except for the descriptions regarding statistical software - should be in the result section.

Results

  • There is no need to describe what is in the Table one by one in your manuscript. Your results can be shortened. 
  • It is not clear how you defined comorbidity. I guess comorbidities include HTN, DM, asthma, LC, COPD, autoimmune diseases, cerebral diseases, HF, HBV infection, renal diseases, and osteoporosis. It seems to me that you just went over all diseases that these patients were diagnosed with. I don't see why you mentioned all these diseases in the table. If you were to categorize the study subjects with number of comorbidities, was it really necessary to show all these? Why did you focus on the number of comorbidities, or comorbidities as a whole? 
  • Table 1 requires footnote for abbreviations. 
  • Figure 1 should appear prior to Table 2 since its description in the manuscript precedes that of Table 2's. Since I am not familiar with MCA and all these mathematical descriptions, it would be helpful for readers like me if some more descriptions regarding Figure 1, specifically how you can interpret the results, can be added. What's the meaning of the number of classifications? Is it just cluster numbers or is there any priorities regarding the numbers?
  • I guess Table 2 can be better elaborated. It's not quite distinct at this point. 

Discussion 

  • I am still confused why this novel algorithm is valuable. It can be possible that the reason why the results of logistic regression was not significant may lie on the fact that appropriate covariates were not selected in the first place. Some important factors that might influence development of depression and anxiety, such as family history of mental diseases and severity of cancer progress (stage), should be considered. Then, the results may not be the same.
  • It would be helpful for me to understand your work better if you can explain more about the importance of this novel method. Although you mentioned some in the discussion section, it doesn't quite stand out. 

Minor comments 

English needs proofreading. 

Author Response

(The authors gave the same response as above.)
